# A Cross-Sectional Study to Measure Physical Activity with Accelerometry in ADHD Children according to Presentations

**DOI:** 10.3390/children10010050

**Published:** 2022-12-26

**Authors:** Lorena Villalba-Heredia, Celestino Rodríguez, Zaira Santana, José Carlos Nuñez, Antonio Méndez-Giménez

**Affiliations:** 1Department of Physiatry and Nursing, Faculty of Health Science, University of Zaragoza, 50009 Zaragoza, Spain; 2GENUD (Growth, Exercise, Nutrition and Development) Research Group, Department of Physiatry and Nursing, University of Zaragoza, 50009 Zaragoza, Spain; 3EXERNET, Physical Exercise and Health Research Network for Special Populations, 50009 Zaragoza, Spain; 4Faculty of Psychology, University of Oviedo, 33003 Asturias, Spain; 5Psicotogether, Psychopedagogy Unit, La Paloma Hospital, 35005 Las Palmas de Gran Canaria, Spain; 6Faculty of Social Sciences and Humanities, Universidad Autonoma de Chile, Temuco 4810010, Chile; 7Faculty of Education, University of Oviedo, 33005 Asturias, Spain

**Keywords:** ADHD, accelerometry, physical activity, intensity, ADHD patterns, primary education

## Abstract

(1) Background: Attention deficit hyperactivity disorder (ADHD) is a common mental disorder affecting 5–7% of school-aged children. Previous studies have looked at the effects of physical activity interventions on the symptoms of ADHD, although few have compared the motor behavior of children with ADHD versus those without. This exploratory study provides detailed information on the patterns and intensity of physical activity and sedentary behavior in children with ADHD as measured by Actigraph GT3X accelerometry, as well as the differences in physical activity in the different presentations of ADHD; (2) Methods: A cross-sectional design was used with a sample of 75 children, aged 6 to 12 years, with and without ADHD. The ADHD group had a previous diagnosis, determined by clinical assessment based on DSM-5 criteria; (3) Results: Physical activity levels were higher in children with ADHD compared to children without ADHD, but there was no difference in sedentary time between groups during weekdays or weekends. Physical activity decreased with age, with significant differences in the ADHD group, who exhibited more minutes of moderate Physical activity in 6–7 year-olds than 10–11 year-olds during weekdays and weekends; (4) Conclusions: Sedentary time increased by age in children without ADHD, and there was a decrease in moderate-intensity physical activity time in children with ADHD by age.

## 1. Introduction

Attention Deficit Hyperactivity Disorder (ADHD) is one of the most common neurodevelopmental disorders, with an incidence of 5–7% of the school-age population, and is more common in boys than in girls [1]. In Spain, the prevalence is 6.8% [2]. ADHD, with childhood-onset that in most cases persists into adolescence and adulthood, corresponds to a symptomatic triad, marked by a combination of signs or patterns of steady indicators such as hyperactivity, impulsivity, and attention deficit which can have negative impacts on household, school, social, and occupational functioning [3]. In addition, the DSM 5 lists 18 symptoms in two dimensions: hyperactivity-impulsivity and inattention. Depending on the presenting symptoms, there are three main possible presentations: predominantly hyperactive/impulsive presentation, predominantly inattentive presentation, and combined presentation. For a diagnosis, at least six signs of hyperactivity-impulsivity and/or inattention must be present in infancy and young adult life, and a minimum of five signs in adult life must be visible in two or more environments (e.g., family, at school, after-school tasks, at work, with friends) for at least six months [3].

According to Sempere-Tortosa et al. [4], there are still many questions about ADHD’s etiology, prevalence, assessment instruments, and procedures. The main reason for these issues lies in the disagreement in establishing diagnostic criteria. In the lack of any biochemical, structural, or inherited factors to unequivocally establish a diagnosis of ADHD, the assessment is clinical, i.e., it is determined by the professional judgment of the physician, and it is based on parents’ and teachers’ observations and information [5]. Furthermore, as the Spanish Association of Neuropsychiatry has pointed out, “the major weakness lies in the fact that the clinical criteria used to diagnose ADHD are too general and vague. The diagnostic manual used, the DSM-5, includes very extensive and subjective criteria. How is it possible to determine whether a child is becoming susceptible to a greater or lesser level of motion?” [6].

Accordingly, the inexactness of the diagnosis of ADHD—relying on non-objective criteria [7]—coupled with the fact that hyperactivity is the main symptom of this disorder [8,9,10] means that several studies have been conducted to record objective measures of movement in subjects. Sempere-Tortosa et al. [11] published results showing differences in the degree of target movement between an ADHD clinical group and a control group, with the control group exhibiting greater mean movement than the experimental group.

According to the recommendations of the World Health Organization (WHO), children and adolescents aged between 5 and 17 should do an average of at least 60 min of moderate or intense physical activity (MVPA) per day and may expect additional benefits with more activity [12]. Despite that, many children and adolescents do not do the recommended amount of physical activity, with over 80% of the world’s adolescent population being insufficiently physically active in 2016 [13]. In the Spanish population, 61% of school children perform at least 60 min of physical activity outside of school more than 2 days a week, which is not enough according to the WHO recommendations [14].

It is also important to mention sedentary lifestyles. Sedentary time is defined as periods involving energy expenditure ≤1.5 metabolic equivalents (METs) during waking time [15]. Decreasing physical activity and thus increasing sedentary time begins in childhood [16,17]. According to a study coordinated by the Active Healthy Kids Global Alliance, which studied data from 38 countries, more than 60% of children are exposed to more than 2 hours of screen time per day, above the WHO recommendations [18]. Both physical activity and sedentary lifestyles are factors related to executive functions [19], and the motor symptomatology of ADHD can be explained by motor disturbances being a feature of this disorder [20]. Children with ADHD have high movement patterns and low sedentary patterns, ADHD-I being the most likely to exceed the recommendations for sedentary behavior [21]. Along these lines, Zang [22] reported that children with ADHD exhibited impaired sensory processing that could lead to hyperactive behavior, manifested through more physical activity in daily life.

Lin, Yang, and Su [23] showed that children with ADHD had higher levels of AFMV than children without ADHD. The study participants wore a PA monitor for 14 h per day for one week. The children’s daily activity records were completed by the parents. Children with ADHD exhibited higher average PA values than children with normal development during those 7 days. This study supported the hyperactivity of children with ADHD in daily life [24,25]. Sensory modulation problems were detected using an instrument to assess the sensory profile (SP) and by means of a Sensory Challenge Protocol that measured the electrodermal response (EDR). Compared to controls, ADHD children reported a higher overall PA level on weekdays (>35%) and weekends (>57%). Nevertheless, when analyzing recorded PA by time, cluster differences were apparent only for some hours. Our findings suggest that ADHD children were hyperactive more in non-structured environments than in structured environments. The ADHD group showed their sensory modulation problems in the SP rather than in the EDR. The hyperactivity distribution appeared to be influenced by social and temporal contexts, underscoring the importance of properly organizing the daily activities of ADHD children. However, most previous studies have been interventions focused on increasing levels of moderate or vigorous physical activity with a range of activities that require different physical and motor skills rather than measuring the amount of physical activity on a typical day [26,27]. Along these lines, other authors, such as Villalba-Heredia et al. [28], assessed the relationship between physical activity, sleep, and academic performance, reporting results indicating that ADHD has an effect on the amount of sleep during non-school days, waking time at the weekend, sleep efficiency during the week, and academic performance. On the other hand, authors such as Best [29] have stated that aerobic exercise seems to improve executive function, and it is possible that activity requiring greater cognitive commitment may have a greater impact on executive function than simpler exercises that require limited cognitive involvement. This is similar to a study from Michigan State University [30], where they concluded that a few minutes of exercise could help ADHD children to improve their academic performance. The study showed for the first time that children with ADHD were better able to suppress or manage distractions and better at concentrating on a task after an exercise session. These and other authors reinforce the importance of observing PA and sedentary time in relation to improving executive functions and academic performance [31]. Motor deficits are very common in children with ADHD, with a prevalence of 30% to 50% of children with the disorder [32]. According to some studies, locomotor hyperactivity is a feature of ADHD that remains into adulthood, with these motor control deficits being more prevalent in children aged 6 to 9 and seeming to attenuate with increasing age [33]. Recently, a study with an African child population looked at the movement patterns of children with ADHD and found mean values of approximately 20 min/day of moderate physical activity, 4 min/day of vigorous physical activity, and 296 min/day of sedentary activity [34].

Measurement of physical activity is as important as it is complex, and there are many ways it can be quantified. As Aparicio-Ugarriza et al. [35] noted, recent years have seen the development of various objective measurement devices, such as accelerometers, that are appropriate for field studies, in contrast to subjective methods such as questionnaires and physical activity diaries [36]. The present study is a pilot cross-sectional study that examined the amount and intensity of physical activity and the degree of sedentary lifestyles in children with ADHD using accelerometry, assessing the different presentations and making a comparison to children without ADHD. It may provide data that will help to increase knowledge about the characteristics of ADHD children. Understanding patterns of PA in children with ADHD may help us better understand the diagnosis in relation to movement.

## 2. Materials and Methods

### 2.1. Participants

The sample comprised 75 children aged 6 to 12 years old, with an average age of 9.47 (SD = 1.73), from public and private elementary schools in northern Spain. A control group of 28 children without ADHD and a group of 47 ADHD children were included.

### 2.2. Procedure and Exclusion/Inclusion Criteria

The study complied with the ethical regulations of the Declaration of Helsinki [37] and was approved by the Ethics Committee of the Principality of Asturias (CPMP/ICH/135/95, code: TDAH-Oviedo). Participation in the study was voluntary, and the anonymity and ethical treatment of the data were ensured. The STROBE guidelines for Strengthening the Reporting of Observational Studies in Epidemiology were used in this study.

The sample was split into two groups: the “ADHD” group of students diagnosed with ADHD, attending state-funded and private schools in the Principality of Asturias, and the “Non-ADHD” group of students in a state school. The sample of children with ADHD was provided by two different hospitals dealing with ADHD in a single health authority region. The evaluation was carried out by a clinician, with the families’ approval, in order to verify the inclusion of the participants in the study. The participants came from different schools selected for ease of accessibility, all from the same area of the Principality of Asturias, and all following the same teaching system. There were no differences between the schools since Spanish schools follow a state curriculum, with a school schedule of 25 h per week and the same number of hours of physical education for all participants—2 h per week. Schools were not analyzed separately to determine whether there were differences in intensity between them.

To be included in the study, subjects in the ADHD group had to meet the following requirements (based on the DSM-5 criteria [3]): (1) must exhibit an impulsive pattern of inattention and/or hyperactivity lasting at least 6 months; (2) inattention and/or hyperactivity-impulsivity symptoms must impair performance or developmental levels; (3) these symptoms must be apparent before the subject is 12 years old; (4) multiple symptoms must be present in at least two or more contexts (e.g., home or school, with friends or family, or other activities); (5) there must be clear evidence that symptoms impair or impair functioning in social, school, or work settings. Additionally, participants must not be receiving any treatment. Data collection was conducted prior to the COVID-19 lockdown between March 2019 and June 2019.

In order to define the groups and to determine the relationship between the variables in relation to the different presentations of ADHD, clinicians confirmed the DSM-5 criteria based on the results from the Farré and Narbona test [38], Evaluation of Attention Deficit Hyperactivity Disorder (ADHD). The “Inattentive” group was made up of students who scored above the 90th percentile in the AD (attention deficit) items. The “Hyperactive/Impulsive” group was made up of students with the Hyperactive/Impulsive presentation, scoring above the 90th percentile in the H (Hyperactivity/Impulsivity) items. The “Combined” group, with the presentation combining Attention Deficit and Hyperactivity, were those scoring above the 90th percentile in the total of the items from the H + AD (Hyperactivity/Impulsivity and Inattention) items.

### 2.3. Instruments

A variety of data collection tools were used in the study based on the variables to be included.

#### 2.3.1. Sex, Age, Socioeconomic Level, School Year, and Type of School

This information was provided by the school, the parents, or the students.

#### 2.3.2. Physical Activity (PA) and Sedentary Time

The physical activity level of the students was objectively measured with ActiGraph-GT3X accelerometers (ActiGraphTM, LLC, Fort Walton Beach, FL, USA) [39]. Participating students wore accelerometers 24 hours a day for a week (24/7). Participants were instructed to wear the accelerometer on the right hip, secured by an elastic belt, at all times. As the monitors were not waterproof, participants were asked to remove them during showers or baths or if swimming or diving; these times were considered non-wearing times. The sampling interval (epoch) was set at 10 s [40]. The minimum number of days to be included in the sample was 5 days, including at least one weekend day [41]. Non-wear was defined as any sequence of 10 min of consecutive zero counts per minute [36]. Data were collected during the second quarter of the academic year. Using these accelerometers made it possible to objectively quantify both the frequency and intensity of the participant’s physical activity (light, moderate, and vigorous), in addition to providing information about rest time (hours of sleep) and hours of inactivity (sedentary), which is one of the main strengths of this study. The sleep variable was collected and analyzed in a previous study [28]. Data were quantified using the cut-off points established by the Freedson et al. [42] algorithm for 6–12 year-olds, provided as part of the ActiLife software. The frequency and intensity of physical activity during one week were objectively measured and recorded for 24 h a day. The accelerometers were placed on the hip with a flexible band since this is the location that has been proven to provide the most reliable data [43].

The categories of intensity are given in Table 1, based on steps per minute, also known as counts per minute (cpm) [42].

#### 2.3.3. ADHD

The Assessment of Attention Deficit with Hyperactivity Scale (EDAH) by Farré and Narbona [38] measures the main characteristics of ADHD and behavioral disorders coexisting with ADHD. The scale consists of 20 items, 5 for Attention Deficit, 5 for Hyperactivity/Impulsivity, and 10 for Behavior Disorders. The questionnaire is completed by students’ parents or teachers. Responses are given using a 4-point Likert format with values ranging from 0 to 3. Cronbach’s alpha was high for the full scale (α = 0.929) and the subscales: AD (α = 0.898), H (α = 0.849), and CD (α = 0.899). For the purposes of the present study, only the AD and H subscales were considered. Scores above 90% indicate inattention deficit and/or hyperactivity-impulsivity. In the present study, a high-reliability index was obtained, α = 0.97.

### 2.4. Statistical Analysis

All of the collected data was stored in an SPSS 25.0 database. To analyze whether there were statistically significant differences between the groups, univariate and multivariate analyses of covariance (ANCOVA and MANCOVA (with covariates age and sex)) were performed. The independent variable was first the group (ADHD and control) and then the ADHD presentation groups (as established by the “Statistical Manual of Mental Disorders” DSM-5 [3]). The dependent variables were the scores in the physical activity variables (weekday and weekend counts, minutes of sedentary activity, and light, moderate, or vigorous physical activity on weekdays and weekends). To determine between-group differences, we used the multiple comparisons test with the Scheffé test. The sample followed a normal distribution according to Gravetter and Walnau [44], with the exception of vigorous PA). An indicator of the magnitude of the effect [45] was included: Cohen’s (1988) delta was used as a criterion, according to which the effect is small when ηp^2^ = 0.01 (d = 0.20), medium when ηp^2^ = 0.059 (d = 0.50), and high when ηp^2^ = 0.138 (d = 0.80).

## 3. Results

### 3.1. Preliminary Results

The descriptive body composition and socio-educational variables of the 75 children participating in the study are shown in Table 2. No significant differences between groups were found by age (*p* = 0.231) or BMI (*p* = 0.161). The groups were not offset by sex since ADHD is more frequent in boys than in girls (χ2(1) = 4.587); (*p* = 0.032). Table 2 provides the main descriptive statistics for the sample in relation to the variables analyzed in this study.

### 3.2. Relationship between Counts, Minutes of Physical Activity Intensity, Sedentary Minutes and ADHD

The variables sex and age in the MANCOVA analysis were not statistically significant as covariates (*p* = 0.703), where nonparametric tests were used in vigorous PA (Table 3. The ADHD group had higher weekday counts than the non-ADHD group, with the difference being statistically significant (F(1.75) = 27.514; *p* < 0.000; d = 1.25). The ADHD group also spent more minutes during weekdays than the non-ADHD group in light PA (F(1.75) = 35.994; *p* < 0.000; d = 1.43), moderate PA (F(1.75) = 43.986; *p* < 0.000; d = 1.58;) and vigorous PA (F(1.75) = 16.983; *p* < 0.000; d = 0.98;). There were statistically significant differences in sedentary minutes between the groups during weekdays (F(1.75) = 7.398; *p* < 0.01; d = 0.65), with higher values of sedentary behavior in children with ADHD than in the group without ADHD. Medication had no statistically significant effects on the weekday variables *p* = 0.952.

Similarly, during the weekends, the ADHD group had higher counts than the Non-ADHD group, with the differences being statistically significant (F(1.75) = 20.085; *p* < 0.000; d = 1.07). The ADHD group also spent more weekend minutes than the Non-ADHD group in light PA (F(1.75) = 21.129; *p* < 0.000; d = 1.10), moderate PA (F(1.75) = 32.599; *p* < 0.000; d = 1.36), and vigorous PA (F(1.75) = 25.026; *p* < 0.000; d = 1.19). There were also statistically significant differences between the groups during weekends in sedentary time (F(1.75) = 6.939; *p* < 0.05; d = 0.63) with more minutes of sedentary time in the ADHD group than the Non-ADHD group. As with weekdays, medication had no statistically significant effect on weekends *p* = 0.667.

Differences between the sexes were found for weekend PA variables, with significantly higher values for boys in weekend vigorous PA (F(1.75) = 4.851; *p* < 0.05; d = 3.25). Likewise, differences between the sexes were found in the ADHD group weekday PA variables, with significantly higher values for boys in weekday counts (F(1.47) = 10.9; *p* < 0.01; d = 5.13) and moderate weekday PA (F(1.47) = 4.969; *p* < 0.05; d = 3.46). In the Non-ADHD group, there were differences between the sexes in weekday and weekend PA variables, with significantly higher values for girls in weekday counts (F(1.28) = 4.476; *p* < 0.05; d = 0.80), moderate PA (F(1.28) = 8.051; *p* < 0.01; d = 1.08), light PA (F(1.28) = 9.818; *p* < 0.01; d = 1.19) and sedentary time (F(1.28) = 4.388; *p* < 0.05; d = 0.79).

Similarly, during the weekends, there were significant differences in counts (F(1.28) = 4.644; *p* < 0.05; d = 0.82), moderate PA (F(1.28) = 6.39; *p* < 0.05; d = 0.96), light PA (F(1.28) = 12.145; *p* < 0.01; d = 1.32) and sedentary time (F(1.28) = 7.248; *p* < 0.05; d = 0.37) with higher values for girls. In the Non-ADHD group, no differences were found for either weekday or weekend vigorous PA.

### 3.3. Physical Activity Parameters and the Different Presentations of ADHD

Table 4 shows the ANOVA results of the interaction between weekend and weekday physical activity and the different ADHD presentations.

Both during the week and at weekends, the hyperactive/impulsive and combined groups of students had statistically significantly higher counts than the control group. They also spent statistically significantly more time than the control group in light, moderate, and vigorous physical activity, both during the week and at weekends. The only statistically significant difference between the control group and the inattentive group was in the amounts of moderate physical activity, both during the week and at weekends; the inattentive group had higher values than the control group. There were statistically significant differences in the weekday count means of the control group compared to the Hyperactive/Impulsive and Combined groups (*p* = 0.002, *p* = 0.000), as well as in the weekend count means of the control group compared to the Hyperactive/Impulsive and Combined groups (*p* = 0.002, *p* = 0.009), the control group exhibited lower values in both cases.

There were also similar differences in the mean amount of light physical activity over a weekday between the control, Hyperactive/Impulsive, and Combined groups (*p* = 0.000, *p* = 0.000) and in the mean amounts of weekend light physical activity between the control, Hyperactive/Impulsive and Combined groups (*p* = 0.005, *p* = 0.000). There were also statistically significant differences in mean weekday minutes of moderate activity between the control group and all three ADHD presentations, with lower values for the control group compared to the Inattention, Hyperactive/Impulsive and Combined groups (*p* = 0.022, *p* = 0.000, *p* = 0.000), with a similar pattern for weekend minutes of moderate activity (Table 4).

A similar pattern was evident in the mean amounts of weekday vigorous activity between the control, Hyperactive/Impulsive, and Combined groups (*p* = 0.004, *p* = 0.002), which was repeated with the mean amounts of weekend vigorous activity between the control group and the Hyperactive/Impulsive and Combined presentations (*p* = 0.016, *p* = 0.002).

Similarly, statistically significant differences were found in mean weekend minutes of vigorous physical activity between the control group and the Hyperactive/Impulsive and Combined presentations (*p* = 0.016, *p* = 0.002), respectively, with higher values for the Hyperactive/Impulsive and Combined groups compared to the non-ADHD group.

There were no statistically significant differences in the means of sedentary time between the groups examined, either during the week or at weekends.

### 3.4. Physical Activity Intensity and Sedentary Behavior Changes by Age

Looking more deeply at the age range using Scheffé’s post hoc test for the ADHD group in Table 5, we found a decrease in the number of minutes of moderate physical activity by age. This difference was statistically significant for both weekdays and weekends in the ADHD group (*p* < 0.05), with significantly more minutes of moderate PA in 6–7 year-olds than 10–11 year-olds (d = 5.37, MD = 412.33, *p* = 0.03) on weekdays.

Similarly, at weekends, 6–7 year-olds exhibited more minutes of moderate PA compared to 10–11 year-olds (d = 3.82, MD = 164.72, *p* = 0.045), and there were higher levels for 8–9 year-olds than 10–11 year-olds (d = 4.23, MD = 154.48, *p* = 0.014). There were no differences between age in the other physical activity and sedentary behavior variables for the ADHD group (*p* > 0.05).

We did find age-related differences in the non-ADHD group for minutes of light physical activity on weekdays (d = 4.53, MD = −328.18, *p* = 0.014) with an increase in minutes in these variables as a function of age. Children aged 8–9 had more sedentary minutes on weekdays than those aged 6–7 (d = 4.81, MD = 1807.31, *p* = 0.008), and children aged 10–11 had more sedentary minutes than those aged 6–7 (d = 6.63, MD = 2421.08, *p* = 0.000). On weekends, sedentary minutes also increased as a function of age, with 10–11 year-olds having more than 6–7 year-olds (d = −4.72, MD = −875.96, *p* = 0.010). There were no differences between ages in the other physical activity variables for the Non-ADHD group (*p* > 0.05).

## 4. Discussion

This study produced valuable findings about the amounts of moderate physical activity in children with and without ADHD, additionally demonstrating whether the participants did the amount of physical activity that the WHO recommends for children. Children with ADHD did more PA than non-ADHD children, but about 92% of the sample in both groups complied with the WHO guidelines for PA in children [12], with a mean of 60 daily minutes of moderate-to-vigorous intensity, mainly aerobic, physical activity during the week. The amount of sedentary time did not seem to correlate with ADHD or any of the presentations of ADHD. The findings of the current study are important because we identified associations that help us to understand the importance of ADHD presentations when it comes to higher levels of physical activity. We did not really know the relationship between ADHD and the different intensities of physical activity measured by accelerometry previously due to a lack of studies on the subject, although it is important to highlight authors such as Neudecker et al. [46] who suggest that PA may help to decrease the negative effects of ADHD contributing to long-term health benefits in children and adolescents with ADHD.

In this sample of children with and without ADHD, there was no difference in the objective measurements of physical activity between boys and girls, although differences were found within the non-ADHD group by sex. Our results are consistent with previous studies, which found sex differences in the amount and intensity of PA in non-ADHD children [47]. We did, however, see notable differences in the frequency and intensity of physical activity that could be attributed to the presence of ADHD. A previous study using a child population aged 6 to 11 years, with PA monitoring for 14 h a day, seven days a week, found higher levels of vigorous and moderate physical activity in children with ADHD compared to those without ADHD [23].

Sedentary time has been shown to be more important than moderate or high activity in predicting ADHD symptoms as a diagnostic measure of activity [48]. However, we saw no significant differences in sedentary patterns. Boonstra et al. [49] reported that sedentary time and sedentary prevalence did not differ between children and adolescents with and without ADHD, whereas higher physical activity patterns have been found in children with ADHD compared to children without ADHD [50]. Various studies have provided supporting evidence for this line of thinking, and this variable could be used as a predictor of ADHD. In terms of the amounts of physical activity in relation to the different presentations of ADHD, we found higher activity in the hyperactive/impulsive and combined presentations compared to the control group, in line with at least one other study [48]. In contrast, other studies have not found any differences in physical activity patterns based on ADHD presentations [51].

Other studies using objective measurements to analyze movement, via infrared [33,52,53] or Kinect, with the ADHD Movements program [4,11] also reported greater physical activity in children with ADHD than children without. Sempere-Tortosa et al. [4] showed a higher concentration of activity in the afternoon compared to the morning, when no differences between groups were observed. Our study showed higher activity during weekends than during weekdays, something that could be explained by previous research, given that children are at school during weekday mornings and the highest motor activity occurs in the afternoons. On weekdays, students are at school in the morning, mostly sitting down, except for physical education. This reason, as well as the fact that in Spain, some after-school sports competitions are on weekends, may explain why there is more physical activity during weekends.

In line with Volkmar’s [25] results, children with ADHD had a generally higher level of PA than the control group and tended to spend more time in moderate PA on weekdays and at weekends. Our data suggested that children with ADHD were more hyperactive at weekends than on weekdays.

Previous research has reported similar mean effect sizes in children and in adolescents [33], although some studies have reported differences by age [54]. In line with Kofler et al. [54], in our study, we found differences in the ADHD group in moderate physical activity, with a decrease in moderate physical activity as children got older. On the other hand, we found no differences in this group in sedentary time by age, which may be due to the motor load that is partly explained by the disorder itself, along with other variables that were not studied. In line with Kontostoli et al. [55], which showed how sedentary time increases by 21 min each year starting in childhood, our study indicated age-related differences in sedentary time and light physical activity in children without ADHD. Our study is consistent with the recent study by Santiago-Rodriguez et al. [34], which reported similar values for sedentary physical activity, although our sample exhibited slightly lower values in moderate and vigorous physical activity than African children of the same age with ADHD. Regarding age, similarly to Murillo et al. [33], the levels of moderate and vigorous physical activity decreased as children aged. There are no cut-off points or algorithms for relating children’s physical activity to ADHD or ADHD presentations. There is evidence that the amount and intensity of physical activity differ between children with and without ADHD [56], using the limited research to date with accelerometry as an identifier of ADHD through activity [8].

This study has some limitations. It did not consider other comorbid disorders, nor were the use of stimulant treatments or their effects on ADHD subjects’ locomotor activity examined. The main limitation of our study was the sample size of the subgroups for the different presentations of ADHD. A lack of previous studies providing information on amounts of different intensities of activity or activity related to ADHD presentations on both weekdays and weekends makes it difficult to compare the patterns and motor behavior based on the intensity of physical activity of children with ADHD versus those without ADHD. It would be interesting to record the intensity of physical education classes, as well as to record whether children engage in active transportation. Other limitations of the study are the size and accessibility of the sample—our sample size was small, and it was a pilot study—as well as not examining the influence of variables such as ethnicity/race or socioeconomic status. ADHD is a disorder with a higher prevalence in boys than in girls, which may imply a limitation in the homogeneity of the sample. One of the strengths of this study is the use of GT3X accelerometry to measure physical activity and sedentary time patterns in children with and without ADHD. Future studies might find it interesting to determine the diagnostic utility of accelerometry in patients with this disorder, looking to see whether there is variability when analyzing other ethnicities and lifestyles and the interaction of age and sex on motor symptoms based on a larger sample.

## 5. Conclusions

In conclusion, greater physical activity was associated with ADHD and ADHD presentations in children with ADHD compared to Non-ADHD children. We found that 92% of our 6–12-year-old population did sufficient levels of physical activity, although children with ADHD had higher levels of MVPA. Children with ADHD did not present differences in the amount or intensity of physical activity by sex, while in children without ADHD, girls demonstrated more activity and a greater intensity of physical activity than boys. We reported an increase in sedentary time by age in Non-ADHD children and a decrease in moderate-intensity physical activity time in ADHD children by age. There is a need for additional work to investigate these topics more fully. A metanalysis might be helpful in terms of conceptualizing how/why increased gross motor movements may manifest in children with ADHD [57].

## Figures and Tables

**Table 1 children-10-00050-t001:** Intensity of physical activity based on counts per minute.

Intensity	Counts per Minute (cpm)
Sedentary	0–149
Light	150–499
Moderate	500–3999
Vigorous	>4000

**Table 2 children-10-00050-t002:** Main descriptive statistics.

	ADHD (*n* = 47)	Non-ADHD (*n* = 28)	ALL (*n* = 75)
**Sex N (%)**			
Male	33 (70.2)	15 (53.6)	48 (64)
Female	14 (29.8)	13 (46.4)	27 (36)
**ADHD Presentations N (%)**			
Inattentive	9(19.1)	-	9 (12)
Hyperactive/impulsive	10 (21.3)	-	10 (13.4)
Combined	28 (59.6)	-	28 (37.3)
Control group	-	28 (100)	28 (37.3)
**Weight M (SD)**	35.83 kg (11.37)	31.93 kg (7.70)	34.37 kg (10.27)
**Height M(SD)**	137 cm (0.11)	135 cm (0.11)	136 cm (0.11)
**BMI M(SD)**	18.61 kg/m^2^ (3.73)	17.40 kg/m^2^ (2.51)	18.16 kg/m^2^ (3.36)
**Age M(SD)**	9.62 years (1.53)	9.21 years (1.95)	9.47 years (1.70)
**School year N (%)**			
Years 1 and 2	9 (19.1)	7 (25)	16 (21.3)
Years 3 and 4	20 (42.6)	10 (35.7)	30 (40)
Years 5 and 6	18 (38.3)	11 (39.3)	29 (38.7)
**Medication N (%)**			
Yes	30 (63.8)	-	30 (40)
No	17 (36.2)	28 (100)	45 (60)
**School type N (%)**			
State-funded	26 (55.3)	28 (100)	54 (72)
Private	21 (44.7)	-	21 (28)
**Extracurricular sports activities N (%)**			
0 h/week	10 (21.3)	8 (28.6)	18 (24)
2 h/week	14 (29.8)	3 (10.7)	17 (22.7)
3 h/week	10 (21.3)	-	10 (13.3)
5 h/week	6 (12.8)	7 (25)	13 (17.3)
7 h/week	7 (14.8)	8 (28.6)	15 (20)
More than 7 h/week	-	2 (7.1)	2 (2.7)

M = Mean; SD = Standard Deviation; BMI = Body Mass Index.

**Table 3 children-10-00050-t003:** Inter-subject effects of ADHD on Physical Activity variables.

	Asymmetry	Kurtosis	ADHD M (SD)	Non-ADHD M (SD)	*F*	*p*	*ηp^2^*
**Weekdays**							
Counts	0.331	0.339	59288.15 (19348.28)	35727.71 (17869.61)	27.51	0.000	0.274
Minutes of light physical activity	−1.215	1.964	793.38 (163.15)	508.68 (247.96)	35.99	0.000	0.33
Minutes of moderate physical activity	−0.250	0.228	1159.83 (317.4)	660 (312.69)	43.99	0.000	0.376
Minutes of vigorous physical activity	2.738	10.598	49.89 (51.96)	6.43 (25.91)	16.98	0.000	0.189
Sedentary minutes	−1.344	1.956	3667.21 (729.11)	2985.89 (1439.15)	7.40	0.008	0.092
**Weekend**							
Counts	0.331	0.159	22700.53 (9739.73)	12653.21 (8765.11)	20.09	0.000	0.216
Minutes of light physical activity	−1.019	0.588	337.5 (80.81)	217.71 (145.23)	21.13	0.000	0.224
Minutes of moderate physical activity	−0.137	−0.027	494.45 (174.01)	263.21 (161.93)	32.60	0.000	0.309
Minutes of vigorous physical activity	1.622	1.861	19.68 (20.17)	0.50 (1.92)	25.03	0.000	0.255
Sedentary minutes	−1.323	1.388	1402.44 (336.81)	1103.68 (645.73)	6.94	0.010	0.087

Weekday counts and minutes are the totals over five days, and weekend counts and minutes are the totals over two days.

**Table 4 children-10-00050-t004:** Inter-subject effects for the four groups on physical activity variables.

	AD vs. CG		H vs. CG		H+AD vs. CG		AD vs. H		H+AD vs. AD		H+AD vs. H	
	MD	d	MD	d	MD	d	MD	d	MD	d	MD	d
1	12726.79	0.73	25808.95 ***	1.47	26814.61 ***	1.39	−13082.17	0.79	14087.82	0.72	1005.65	0.05
2	204.12	0.82	312.48 ***	1.42	303.60 ***	1.52	−108.37	0.56	99.48	0.58	−8.89	0.07
3	327.10	0.99	580.50 ***	1.99	530.20 *	1.70	−253.40	0.82	203.10	0.61	−50.3	0.17
4	27.07	0.89	53.82 **	1.13	45.05 ***	1.37	−26.75	0.42	17.98	0.45	−8.77	0.16
5	602.01	0.45	632.61	0.49	736.43	0.65	−30.6	0.00	134.42	0.33	103.82	0.38
6	8773.19	0.90	12954.29 ***	1.46	9161.63 ***	1.03	−4181.10	0.39	388.44	0.04	−3792.66	0.42
7	76.43	0.56	132.04 *	1.03	131.25 ***	1.13	−55.61	0.63	54.81	0.67	−0.79	0.01
8	184.68 *	0.96	314.87 ***	2.09	209.71 ***	1.36	−130.19	0.67	25.03	0.14	−105.16	0.77
9	18.67 *	1.80	21.58 ***	1.43	18.22 ***	1.62	−2.90	0.12	−0.46	0.03	−3.36	0.16
10	239.19	0.39	238.65	0.42	351.44	0.69	0.54	0.04	112.25	0.19	112.79	0.15

MD = Difference in means; d = effect size; AD= Inattentive presentation; H = Hyperactive/impulsive presentation; CG = Control Group; H + AD = Combined presentation; 1. Weekday counts; 2. Weekday minutes of light physical activity; 3. Weekday minutes of moderate physical activity; 4. Weekday minutes of vigorous physical activity; 5. Weekday sedentary minutes; 6. Weekend counts; 7. Weekend minutes of light physical activity; 8. Weekend minutes of moderate physical activity; 9. Weekend minutes of vigorous physical activity; 10. Weekend sedentary minutes. * *p* < 0.05, ** *p* < 0.01; *** *p* < 0.005. Bonferroni’s correction = 0.008.

**Table 5 children-10-00050-t005:** Inter-subject effects of age on Physical Activity variables.

Weekdays		6–7 y	8–9 y	10–11 y	F	*p*	*ηp* ^2^
Counts	ADHD	62935.89	62890.1	53462.11	1.342	0.272	0.058
	Non-ADHD	23482.43	37254.1	42132.55	2.684	0.088	0.177
Minutes of light physical activity	ADHD	889.22	786.8	752.78	2.241	0.118	0.092
	Non-ADHD	293	535.9	621.18	4.970	**0.015**	0.284
Minutes of moderate physical activity	ADHD	1399	1208.05	986.67	6.854	**0.003**	0.238
	Non-ADHD	504	687.4	734.36	1.243	0.306	0.090
Minutes of vigorous physical activity	ADHD	82.11	49.2	34.56	2.698	0.078	0.109
	Non-ADHD	25.71	0	0	2.960	0.070	0.191
Sedentary minutes	ADHD	3373.78	3632.9	3852.06	1.35	0.27	0.058
	Non-ADHD	1389.29	3196.6	3810.36	10.680	**0.000**	0.461
**Weekend**							
Counts	ADHD	22701.22	25244.15	19873.94	1.469	0.241	0.063
	Non-ADHD	6132	14295.9	15309.73	3.008	0.068	0.194
Minutes of light physical activity	ADHD	363.78	341.47	319.94	0.922	0.405	0.04
	Non-ADHD	138.43	241.7	246.36	1.439	0.256	0.103
Minutes of moderate physical activity	ADHD	561.89	551.65	397.17	5.454	**0.008**	0.199
	Non-ADHD	169.43	304.6	285.27	1.684	0.206	0.119
Minutes of vigorous physical activity	ADHD	29	21.59	12.89	2.176	0.126	0.09
	Non-ADHD	2	0	0	3.365	0.051	0.212
Sedentary minutes	ADHD	1315.44	1345.34	1509.39	1.53	0.228	0.065
	Non-ADHD	508.86	1210.8	1384.82	5.548	**0.010**	0.307

## Data Availability

Data availability by email to author correspondence.

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
