# Peer review of "A Cross-Sectional Study to Measure Physical Activity with Accelerometry in ADHD Children according to Presentations"

_children, 2022, doi:10.3390/children10010050_

Round 1

Reviewer 1 Report

1) In the abstract, the authors have talked more about background of the study and less about the results. The results should be more in detail in the abstract.

2) In the introduction, the authors have focused more on ADHD disorder and its diagnosis, as well as their movement deficiencies, and less has been said about the physical activity of ADHD children. I suggest to talk about the previous studies on the physical activity of ADHD children and express the limitations of these studies. Are age and sex differences of your goals? Please explain.

3) Another important point is that the authors mentioned less about the importance of this research as well as the innovation of the research. In fact, the necessity and innovation of the research is not so clear. I suggest that the authors reconsider this matter.

4) Because of limitation in sample size, I think this study is a pilot study.

5) Please explain about the formula of sample size selection.

6) References could be improved buy adding more recent and relevant studies.

Good luck.

Author Response

Review of children-2020113, " A cross-sectional study to measure physical activity with accelerometry in ADHD children according to presentations."

The authors acknowledge the reviewers their suggestions and comments on the manuscript, and hope the changes we introduced will meet their expectations. These changes are described below, in response to the different aspects addressed in the revision.

The modifications in the paper have been highlighted in red as Track Changes in the manuscript.

Review 1

1) In the abstract, the authors have talked more about background of the study and less about the results. The results should be more in detail in the abstract.

  • Thank you, the suggestion has been included the results of the study with more detail in the abstract.

2) In the introduction, the authors have focused more on ADHD disorder and its diagnosis, as well as their movement deficiencies, and less has been said about the physical activity of ADHD children. I suggest to talk about the previous studies on the physical activity of ADHD children and express the limitations of these studies. Are age and sex differences of your goals? Please explain.

  • In accordance with the recommendation, citation [32-34] has been added in allusion to this aspect, extending and justifying the study of the physical activity of ADHD children.

3) Another important point is that the authors mentioned less about the importance of this research as well as the innovation of the research. In fact, the necessity and innovation of the research is not so clear. I suggest that the authors reconsider this matter.

  • Thank you for the appreciation, according to your suggestion, the importance of our study has been included in the discussion, highlighting as one of the strengths is the use of GT3X accelerometry to measure physical activity and sedentary time patterns in children with and without ADHD in Spanish children.

4) Because of limitation in sample size, I think this study is a pilot study.

  • Changes have been made as suggested, thank you, according to it line 132 and 435 has been rewritten, and this information has been included.

5) Please explain about the formula of sample size selection.

  • According to the data obtained by the Instituto Asturiano de Estadística during the 2020-21 academic year, there is a total of 46,971 students enrolled in primary education in Asturias, who meet the inclusion criteria of being between 6 and 12 years old.

To calculate the sample needed for this study, the following formula has been used to calculate the size of the sample size in finite populations:

Where:

N = Population size = 46,971

Z = Confidence level at 95% = 1.96

P = Expected proportion = 0.05

q = Complement of P (1 - P) = 0.95

E = Precision or margin of error = 0.03

n = Sample size

According to this formula, the necessary representative sample was 201 schoolchildren. Therefore, the sample of this study would be composed of a minimum of 201 scholars between 6 and 12 years of age, with or without ADHD, who do not present other comorbid diseases of ADHD, or mobility diseases that impede Physical Activity and/or influence the results. For their selection, accessibility sampling was applied. Given that ADHD is a disorder more frequent in boys than in girls, it was not possible to maintain the proportions of the population in terms of gender, age, type of school and residence in the sample.

Given that the required sample size was not reached, it was proposed as a pilot study to determine the motor and physical activity behaviors of children with ADHD versus non-ADHD school-aged children between 6 and 12 years old.

6) References could be improved buy adding more recent and relevant studies.

  • Thank you, two references of the last year has been added in relation to the suggestion [32,34].

Reviewer 2 Report

The authors recruited 75 children (47 with ADHD and 28 without ADHD) to compare motor behaviors between ADHD cases and controls. They found that greater physical activity was associated with ADHD and the ADHD presentations in children with ADHD compared with Non-ADHD children. Overall it is a well-written article. However, the novelty of the study might be limited, since hyperactivity is one of the core symptoms and diagnostic criteria of ADHD. The second, sample size of this study was small, and gender distribution was unbalanced. The authors need to investigate the interaction of age and sex on motor symptoms based on a larger sample.

Author Response

Review of children-2020113, " A cross-sectional study to measure physical activity with accelerometry in ADHD children according to presentations."

The authors acknowledge the reviewers their suggestions and comments on the manuscript, and hope the changes we introduced will meet their expectations. These changes are described below, in response to the different aspects addressed in the revision.

The modifications in the paper have been highlighted in red as Track Changes in the manuscript.

Review 2

The authors recruited 75 children (47 with ADHD and 28 without ADHD) to compare motor behaviors between ADHD cases and controls. They found that greater physical activity was associated with ADHD and the ADHD presentations in children with ADHD compared with Non-ADHD children. Overall it is a well-written article. However, the novelty of the study might be limited, since hyperactivity is one of the core symptoms and diagnostic criteria of ADHD. The second, sample size of this study was small, and gender distribution was unbalanced. The authors need to investigate the interaction of age and sex on motor symptoms based on a larger sample.

  • Thank you for your appreciation, in accordance with the reply to reviewer 1, we have included in the text that given the small sample size this is a pilot study and we have included this as a limitation of the study. We have also included the suggestion in future perspective to study the interaction of age and sex on motor symptoms based on a larger sample.

Round 2

Reviewer 2 Report

I have no further comment.

Author Response

Thanks for the feedback.
The document has been revised to eliminate what coincides with other manuscripts (if possible). Likewise, the latest version was reviewed by an English language expert.